# A Reproducible Deep-Learning-Based Computer-Aided Diagnosis Tool for Frontotemporal Dementia Using MONAI and Clinica Frameworks

**DOI:** 10.3390/life12070947

**Published:** 2022-06-23

**Authors:** Andrea Termine, Carlo Fabrizio, Carlo Caltagirone, Laura Petrosini

**Affiliations:** 1Data Science Unit, IRCCS Santa Lucia Foundation, 00143 Rome, Italy; a.termine@hsantalucia.it (A.T.); c.fabrizio@hsantalucia.it (C.F.); 2Department of Clinical and Behavioral Neurology, IRCCS Santa Lucia Foundation, 00179 Rome, Italy; c.caltagirone@hsantalucia.it; 3Experimental and Behavioral Neurophysiology, IRCCS Santa Lucia Foundation, 00143 Rome, Italy

**Keywords:** deep learning, computer aided diagnosis, artificial intelligence, MONAI, Clinica, frontotemporal dementia, neurodegenerative diseases, neuroimaging, 3D MRI

## Abstract

Despite Artificial Intelligence (AI) being a leading technology in biomedical research, real-life implementation of AI-based Computer-Aided Diagnosis (CAD) tools into the clinical setting is still remote due to unstandardized practices during development. However, few or no attempts have been made to propose a reproducible CAD development workflow for 3D MRI data. In this paper, we present the development of an easily reproducible and reliable CAD tool using the Clinica and MONAI frameworks that were developed to introduce standardized practices in medical imaging. A Deep Learning (DL) algorithm was trained to detect frontotemporal dementia (FTD) on data from the NIFD database to ensure reproducibility. The DL model yielded 0.80 accuracy (95% confidence intervals: 0.64, 0.91), 1 sensitivity, 0.6 specificity, 0.83 F1-score, and 0.86 AUC, achieving a comparable performance with other FTD classification approaches. Explainable AI methods were applied to understand AI behavior and to identify regions of the images where the DL model misbehaves. Attention maps highlighted that its decision was driven by hallmarking brain areas for FTD and helped us to understand how to improve FTD detection. The proposed standardized methodology could be useful for benchmark comparison in FTD classification. AI-based CAD tools should be developed with the goal of standardizing pipelines, as varying pre-processing and training methods, along with the absence of model behavior explanations, negatively impact regulators’ attitudes towards CAD. The adoption of common best practices for neuroimaging data analysis is a step toward fast evaluation of efficacy and safety of CAD and may accelerate the adoption of AI products in the healthcare system.

## 1. Introduction

Computer-Aided Diagnosis (CAD) tools aim to help improve physicians’ performance in disease detection, with the main objective of detecting early pathological signs that humans may fail to find [1]. CAD applications have been developed for numerous medical imaging modalities and diseases [1,2] and they provide diagnosis probabilities by analyzing the data. As such, CAD tools would help the radiologists to draw their conclusions, supporting their interpretation and decision-making processes [3]. Interestingly, it has been shown that the performance of the radiologist can be equalized by CADs and even improved by joining human collective reasoning with machine predictions [2,4].

CAD on medical imaging data relies on image processing methods and Artificial Intelligence (AI) classification systems. Deep Learning (DL) is the best suited AI technique for this purpose, strong in its role in computer vision. Highly difficult tasks requiring human or superhuman ability have been solved by DL over the last few years, defining new standards in protein structure prediction [5], image generation from natural language description [6], and general-purpose learning in complex domains [7]. Applications of DL methods in the medical field mainly include systems for an earlier or more accurate disease diagnosis, assessment for the risk of conversion to a more severe disease status and disease subtypes identification [8]. DL is based on Artificial Neural Networks that analyze input data by mimicking brain functioning, with several layers of nodes as neurons applying complex transformation functions to data [9]. In the case of a CAD tool, the information is processed and the model produces an output response, such as the prediction of a class probability (e.g., case or control). Implementation and investment in DL for medicine have been growing over the past few years [10]. This is due to its potential of providing new reliable methods to enhance healthcare practices and finally foster precision medicine.

Nonetheless, the acceptance of AI in standard clinical settings is still lagging [11]. One of the factors hindering AI-based CAD spread in hospitals is that DL models have a lack of interpretability, which is a primary concern for medical practitioners. Not knowing how the model made its choice weakens AI trustability. DL models are considered uninterpretable due to the complexity of the transformation they apply when processing the data, and for this reason they are often called black-box models. Recently, the development of explainable AI methods has been tackling this issue. In fact, explainable AI aims to enhance its interpretability by providing insights about models’ behavior [12]. Attention maps are visual tools that help explain deep convolutional neural networks, showing which input regions are the most influential for the network when making a prediction. Producing and interpreting attention maps when developing DL-based CAD tools could strongly enhance their acceptance and finally their diffusion.

Frontotemporal dementia (FTD) is a neurodegenerative clinical syndrome where behavior, executive functions, and language show progressive deficits [13]. FTD ranks third in the prevalence of dementia, after Alzheimer’s disease (AD) and dementia with Lewy bodies [14]. FTD encloses clinical syndromes whose histopathological characteristics are the neuronal loss, gliosis, and progressive neurodegeneration predominantly affecting the frontal and temporal lobes [15,16,17]. Diagnosing FTD is a difficult task and requires a longer period of time compared to AD. This is due to a subtle and insidious onset for FTD, as memory problems, typically the first sign of dementia, are often lacking and the majority of FTD patients have no initial complaints [18,19]. Moreover, FTD is highly heterogeneous in its manifestations and multiple disease variants have been identified, for which international diagnostic criteria have been proposed [20,21,22]. Differential diagnosis of FTD with other forms of dementia such as AD is even harder, and investigations of how long it takes to make a diagnosis of FTD have shown that a comprehensive assessment is essential to differentiate FTD from other diagnoses [23,24]. Even if there is currently no disease-modifying treatment for FTD, its early diagnosis has the potential to improve patient management by timely planning useful pharmacological treatment strategies for symptom control [25]. These are useful in helping caregivers cope with the high levels of distress they are experiencing due to the high prevalence of psychopathology in FTD patients [26]. Understanding how FTD leads to cognitive and behavioral symptoms is critical, and non-invasive brain stimulation methods such as Transcranial Magnetic Stimulation (TMS) and transcranial Direct Current Stimulation (tDCS) have been used to shed light on the mechanisms involved [27,28,29].

Brain imaging in vivo has been crucial to unraveling FTD pathophysiology, as it made it possible to identify several biomarkers associated with the disease. Neuromorphological hallmarks of FTD are gray matter volume loss of the prefrontal cortex, insula and anterior cingulate cortex [30,31]. Additionally, there is a white matter integrity loss over time, and brain atrophy increases with disease progression [32,33]. Magnetic resonance measurements of ventricular volumetric changes in FTD patients showed that their size increases with time, and this correlates with cognitive impairment severity [34]. In particular, ventricular expansion has been found to represent an informative marker to discriminate the behavioral variant of FTD from other FTD variants and from other forms of dementia [35].

The development of a reliable CAD tool needs the data to be carefully annotated, organized, and managed, especially when it is based on Magnetic Resonance Imaging (MRI). Classification systems exploiting brain imaging have been successfully used to capture structural changes in the human brain [36,37] and detect dementia up to flawless performance [38]. Extensive datasets improve the solidity of AI algorithms’ training and many data-sharing initiatives have grown in the field of neurodegenerative disease research in the last 20 years [39]. Such initiatives foster neurodegenerative disease research, by putting aside the need for years-long data collection and providing reliable data. This speeds up both hypothesis testing and data-driven research that exploits AI for data analysis. A key element that enables the use of such techniques is the adoption of academy and industry-wide data standards. In order to make the shared data productive, the FAIR (Findable, Accessible, Interoperable, Reusable) principles have been proposed to promote Open Science practices for data sharing initiatives [40,41]. FAIR data can be highly precious, especially when hundreds of subjects are available and the methods for data acquisition are standardized and reliable. The spread of research-useful data is aided by database management initiatives such as the Image and Data Archive (IDA), hosted by the Laboratory of NeuroImaging (LONI), that contains data from more than 160 studies. IDA shares the Neuroimaging in Frontotemporal Dementia (NIFD) database, which is one the biggest data sharing initiatives on FTD to date. 

To date, few attempts have been made to discriminate FTD from Normal Controls (NC) using AI methods [36,37,42,43,44,45,46,47]. Most of the studies used MRI-derived numerical features, such as gray/white matter volume or cortical thickness quantification, with traditional ML algorithms such as Support Vector Machine (SVM) and logistic regression. To the best of our knowledge, there is only one published study that used 3D MRI data for FTD classification with a pretrained Convolutional Neural Network (CNN) achieving high accuracy [42]. Given that only a few of the previous classification attempts used NIFD data, sample sizes usually ranged from 12 to ~450 subjects, and preprocessing and data augmentation methods, as well as AI algorithms choice and cross-validation strategies, were highly heterogeneous. This heterogeneity in methodology resulted in unstable and potentially unreliable CAD performances, achieving accuracies ranging from 0.66 to 1.

To deepen our understanding of neurodegenerative diseases in general, and FTD in particular, standardized practices are needed when preprocessing and analyzing MRI data with DL, in order to make studies easier and more reproducible. Several initiatives have approached this problem. The Nipype (Neuroimaging in Python—Pipelines and Interfaces) package, an open-source Python project, provides functions where the interaction with neuroimaging software tools or algorithms can be performed in a single workflow, partly handling the heterogeneity of image processing [48]. NiftyTorch is a DL framework for neuroimaging in Python, giving users an interface for PyTorch modeling with the aim of providing a package to easily perform AI-based operations on neuroimaging data [49]. In the same fashion, MedicalTorch is an open-source framework for PyTorch, implementing an extensive set of loaders, pre-processors and datasets for medical imaging [50]. In 2016, efforts to increase data standardization and experiment reproducibility led to the proposal to the neuroimaging community of the Brain Imaging Data Structure (BIDS) as an organization format for clinical and imaging data [51]. Despite the scientific community’s acceptance of the BIDS standard, not all public neuroimaging datasets provide BIDS versions of their data.

Clinica, a set of automatic pipelines for the management of multimodal neuroimaging data, pursues the community effort of reproducibility and aims to make clinical research studies easier [52]. This project is developed and maintained from the Aramis Lab and consists of a set of automatic pipelines for neuroimaging data processing and analysis. The aim of Clinica is to make clinical neuroimaging studies easier and reproducible by providing standardized methods for data processing. Clinica includes converters of public neuroimaging datasets to BIDS, along with processing pipelines, and organization for processed files, statistical analyses, and Machine Learning algorithms.

One of the latest and most extensive initiatives for the standardization of AI applications to medical imaging is “project MONAI”, that originally started in 2020 by NVIDIA and King’s College London and brought to the MONAI framework [53]. It is an open-source PyTorch-based framework for DL in healthcare imaging. The aim of the project is developing and sharing best practices for AI in healthcare imaging, creating state-of-the-art, end-to-end training workflows for healthcare imaging and providing researchers with the optimized and standardized way to create and evaluate Deep Learning models. The MONAI framework provides workflows for using domain optimized networks, loss functions, metrics, and optimizers. 

Considering the high heterogeneity in CAD tools development methodology, there is a need to foster standardized practices to spread the benefits of AI strategies adoption in the diagnosis pipeline. Here, we present a proof-of-concept of Clinica and MONAI application on the NIFD database to train and test a CAD tool on FTD data with the application of shared practices. Our aim was testing the efficacy of only using standardized frameworks for neuroimaging data preprocessing and DL modeling on medical imaging as main steps in CAD development. The achieved performance is comparable to that of other FTD classification systems, showing the appropriateness of this methodology. Explainable AI methods reveal that the model mimics human behavior when making its decision, mainly relying on morphological changes in hallmarking brain areas for FTD. The adoption of academy and industry-wide data standards coupled with standardized practices for neuroimaging data management will provide more reliable results upon the application of less biased procedures. 

## 2. Materials and Methods

### 2.1. Neuroimaging in Frontotemporal Dementia Database

This study was performed on data from the NIFD database, hosted by the Laboratory of NeuroImaging from the University of Southern California. NIFD is the nickname for the frontotemporal lobar degeneration neuroimaging initiative (FTLDNI). FTLDNI was funded through the National Institute of Aging and started in 2010. The primary goals of FTLDNI were to identify neuroimaging modalities and methods of analysis for tracking frontotemporal lobar degeneration (FTLD) and to assess the value of imaging versus other biomarkers in diagnostic roles. The Principal Investigator of NIFD was Dr. Howard Rosen, MD, at the University of California, San Francisco. The data are the result of collaborative efforts at three sites in North America. For up-to-date information on participation and protocol, please visit http://memory.ucsf.edu/research/studies/nifd (accessed on 25 May 2022). NIFD includes data from 346 subjects followed over time including FTD and NC all with a careful assessment through interviews, physical examinations, cognitive testing and blood and/or CSF testing, along with brain MRI and PET acquisition. FTD patients included in this database are diagnosed with one of the following disease variants: behavioral variant, semantic variant, progressive non-fluent aphasia, progressive supranuclear palsy, or cortico-basal syndrome. 

### 2.2. Clinica: An Open-Source Software Platform for Reproducible Clinical Neuroscience Studies

We used Clinica [52] functions to manage NIFD data. In particular, we applied the nifd-to-bids function to convert NIFD data to BIDS format, in order to be ready for processing. Next, we used the t1-linear pipeline to affinely align T1-weighted MR images to the MNI space. With this standardized preprocessing applied, we were ready for model training.

### 2.3. MONAI: Medical Open Network for Artificial Intelligence

We followed the MONAI workflow for 3D classification based on DenseNet. MONAI proposes to use the DenseNet121 architecture stored in PyTorch, a model from [54]. MONAI made importing and transforming the images easy, and provided useful functions to train and test the DL model. Such standardized and community-based practice makes data and model management more solid, increasing the robustness and reproducibility of CAD building practice.

### 2.4. Workflow Overview

The next sections describe the methodology for the proposed workflow, which is shown in Figure 1. NIFD data was downloaded from the IDA and filtered as specified in Section 2.5. Data preprocessing was performed with the standardized pipeline from Clinica, as specified in Section 2.6. Data augmentation was applied only on the train set and preceded model training and test phases, which were all performed in the MONAI framework. These two are described in Section 2.7 and Section 2.8, respectively. Finally, the behavior of resulting CAD is evaluated by extracting attention maps as explained in Section 2.9.

### 2.5. Data Collection

To perform the present study, we filtered the data to include only 3D T1-weighted Magnetization-Prepared Rapid Acquisition with Gradient Echo (MPRAGE) MRI scans at the first visit for cases and controls, as this was the most frequent acquisition modality among many others available in NIFD. Subjects were labeled as cases or controls following the diagnosis reported in the clinical data table. When the diagnosis was missing in the clinical data table, the “patient/control” label available in the MRI metadata was used. This resulted in a final dataset of 182 FTD and 130 NC.

### 2.6. Preprocessing Pipeline

Subjects were randomly assigned to train, validation, and test sets with a 70/30 proportion calculated on the group with the highest n (Table 1). This splitting proportion is most indicated for small sample sizes and it required ~40 subjects for validation and test [55]. Random sampling without replacement resulted in each participant being uniquely assigned to one of the three sets, avoiding data leakage.

The 3D T1 MPRAGE MRI scans underwent BIDS formatting with the application of the nifd-to-bids converter from Clinica [52]. Data preprocessing for the affine registration of T1w images to the MNI standard space was performed using the t1-linear pipeline of Clinica [52,56]. More precisely, bias field correction was applied using the N4ITK method [57]. Next, an affine registration was performed using the SyN algorithm [58] from ANTs [59] to align each image to the MNI space with the ICBM 2009c nonlinear symmetric template [60,61]. The registered images were further cropped to remove the background resulting in images of size 169 × 208 × 179, with 1 mm isotropic voxels.

### 2.7. Data Augmentation Pipeline

MRI scans from subjects in the train set underwent augmentation procedures to generate new observations and enlarge the training set. Data augmentation was performed using MONAI randomized data augmentation transforms. The specs of the applied transformations are described in Table 2. Each image was augmented 5 times.

This process enlarged the train set to 1170 images (715 FTD, 455 NC). In order to build the final train set, all the original images were kept for both groups and the augmented images were sampled to set the n of each group to 400. A summary for the composition of the final train set is shown in Table 3.

### 2.8. Deep Learning Pipeline

A train set of 440 images per group was used to train a classifier following the indications reported in MONAI [53]. A DenseNet121 was trained with default parameters for 3D images for 10 epochs and using a batch size of 2 (Figure 2). The Cross-Entropy loss was used for training with Adam optimizer, setting the learning rate to 1 × 10^−5^. DenseNet121 is a Dense Convolutional Network, where each layer is connected to every other layer and uses all preceding layers’ feature-maps as inputs [54]. When the images were imported in the PyTorch environment they underwent intensity scaling between 0 and 1, a channel was added to make the image in the channel-first format, then it was resized with scaling to 150 the longest dimension, keeping the aspect ratio of the initial image. The model with the best performance on the validation set was saved. Predictions on the test set were performed following MONAI indications and results were evaluated by computing the following evaluation metrics: accuracy, sensitivity, specificity, F1-score, and Area Under the Curve (AUC).

### 2.9. Explainable AI Using the Attention Map Method

In order to investigate where the DL model focused to make its prediction, we applied the Guided Gradient-weighted Class Activation Mapping (Guided Grad-CAM) algorithm using the M3d-CAM tool [62]. M3d-CAM simplifies the interpretability of PyTorch-based models by providing an easy-to-use library for generating attention maps. The application of the Guided Grad-CAM algorithm generated attention maps as new images to visualize important voxels, finally providing insights into model’s behavior [63]. In particular, the neuron importance weights ($\alpha_{n}^{c}$) are computed as the global average pooling of the gradients via backpropagation, before the softmax layer of the DL network. The gradient of the score for the output is computed with respect to feature map activations of a convolutional layer, as shown in Equation (1).
(1)$$\alpha_{n}^{c}=\frac{1}{z}\underset{i}{\sum }\underset{j}{\sum }\underset{k}{\sum }\frac{\partial y^{c}}{\partial A_{ijk}^{n}}\;$$
where *y* is the model output before sigmoid function application, *c* is the class of interest, *A* is the feature map activation, *i*, *j*, and *k* are the width, height, and depth dimensions, respectively. To obtain a tensor heatmap of feature importance ($L_{Grad-CAM}$) representing the activation map of the network Guided Grad-CAM executes a weighted combination of the obtained $\alpha_{n}^{c}$ with $A^{n}$ using the Rectified Linear Unit (ReLU) function, represented in Equation (2) [62].
(2)$$L_{Grad-CAM}=ReLU\;(\underset{n}{\sum }\alpha_{n}^{c}A^{n})$$

The obtained tensor heatmap was then rendered as a pseudocolor image with grayscale colormapping. The areas contributing the most to the model’s output emerged with the greatest variation in grayscale.

## 3. Results

### 3.1. CAD Train and Test 

A DenseNet121 architecture in PyTorch was trained to discriminate between FTD and NC T1 3D MRI following the MONAI framework guidelines. To ensure the model was not performing well only on the data it was trained on, we measured its performance on a separated validation set. In fact, the model was tested on the validation set after each epoch and the model with the best accuracy (0.92) was saved. After training, this saved model was tested on an independent test set to assess prediction performance. The model achieved 0.80 accuracy (95% confidence intervals: 0.64, 0.91), 1 sensitivity, 0.6 specificity, 0.83 F1-score, and an AUC of 0.86. Testing if the accuracy was higher than the no information rate achieved a *p*-value < 0.0001. The confusion matrix is reported in Figure 3, showing that the model only misclassified NC samples, achieving max sensitivity.

### 3.2. Comparison with Previous FTD Classification Approaches

We collected the performance metrics of the studies attempting to discriminate FTD from NC or AD to demonstrate that a reproducible DL powered CAD tool performs similarly (Table 4). Our application achieved 0.80 accuracy, which is in line with the results obtained by other research groups in the FTD vs. NC classification (Accuracy_mean_ = 0.84, sd = 0.08) while showing high reproducibility. Notably, only a few of the previous classification attempts used NIFD data, and their sample size, AI algorithms, and cross-validation strategies were highly heterogeneous. Most of the published DL applications to FTD were designed to discriminate FTD from AD, making it difficult to compare their performance with ours. 

### 3.3. Attention Maps

Attention maps help visually explain DL models for image processing, showing which parts of the image contributed to the classification. We generated attention maps using the Guided Grad-CAM algorithm that provided an image with the same size as the test image, where relevant points have a highly perturbed value, thus emerging from the background and showing where the model focused to perform its prediction. Attention maps for the FTD and the NC subjects with the most accurate prediction are shown in Figure 3. Emerging areas concentrate around the ventricles, where the difference is clearly noticeable, as the FTD subject has expanded ventricles. It also seems that the model partly focuses on the skull. Figure 4 shows a single slice where the difference is evident, yet an animation of the full brain scans and attention maps is available in the Appendix A.

## 4. Discussion

In this work, we showed the use of a standardized workflow to build a CAD tool for FTD. Data preprocessing was made easily reproducible by using the Clinica library [40], while an optimized state-of-the-art DL model was trained within the standardized MONAI framework [53]. The proposed workflow resulted in a CAD with max sensitivity, correctly identifying all FTD samples. Some NC samples were missed, leading to 0.8 accuracy, yet this result is in line with the previous FTD classification approaches based on MRI, whose accuracy in discriminating FTD from controls is around 0.8 [47,68]. Notwithstanding, most of the available papers on FTD classification with AI-based methods used Machine Learning on quantitative variables that, although from MRI brain scans (e.g., cortical thickness), have to be managed differently from 3D MRI data (Table 4). A few works were published where DL is used to detect FTD from 3D imaging, but their focus was on discriminating it from AD, making their results incomparable to ours (Table 4) [42,69]. There is only one published work really comparable to ours [42], but their trained DL model takes raw images as input and uses a non-standard architecture. On such a basis, we found this method built on standardized frameworks to be new in FTD research. Moreover, our work is one of few based on the NIFD database. We believe that our standardized methodology makes our work useful for benchmark comparison in FTD classification. Experiments based on neurodegenerative disease classification need large sample sizes and to the best of our knowledge NIFD offers the largest FTD cohort to date with neuroimaging and clinical data available. Finally, we found that several studies were underpowered, with few participants for each experimental group. Small sample sizes usually lead to unstable results and/or inflated performances of the models used for classification. This bias is worsened when coupled with misused Cross-Validation strategies such as k-fold Cross-Validation that are unreliable with ~20 subjects per group. A standardized workflow following data science best practices protects from data leakage or overfitting, ensuring reproducibility and reliability. 

The performance of the CAD presented here could be further enhanced by training the model with a higher number of epochs, as other works show convergence after 100 epochs [42]. Additionally, tuning the model hyper-parameters such as learning rate and batch size could have yielded higher accuracy, along with choosing a different DL architecture, finally identifying the best performing combination. Moreover, we could have taken a data-centric approach to boost the effectiveness of the training phase, possibly improving test performance. As it has been shown, a data-centric approach counters the challenges of training with a small dataset and improves accuracy [70]. Nonetheless, our approach aimed at simplicity and reproducibility, showing that a CAD tool can be set up by using open-source and easy-to-use software platforms providing state-of-the-art methods for data preprocessing and analysis.

One of the biggest flaws in DL-based CAD tools is their lack of interpretability. In fact, DL neural networks are black boxes, meaning that highly complex data processing makes it unintelligible how the model comes to its final output (a class probability). To the extent of making black boxes more interpretable, a few methods for explaining model behavior have been developed, contributing to the realization of the explainable AI [12,71]. Neurodegenerative disease research is only recently approaching explainable AI [72,73,74,75] and there are only a few works available where Guided Grad-CAM has been used to generate attention maps for DL neural networks [76,77,78,79]. To the best of our knowledge, this is the first work where a DL model for FTD detection is studied with explainable AI methods. As reported in the results section, ventricular spaces were the most influential areas for the model output, and it has also been observed that the model was influenced by skull parts too. The AI presented in this work has a good performance, and attention maps showed that it seems to rely mostly on gross characteristics accounting for large differences in the images, as a human might do. To this extent, this model seems to be reliable as it shows to mimic human behavior when choosing, and this might make this CAD trustable by groups of physicians aiming to find aids from automated analysis methods when evaluating patients. We argue that the model might have misclassified the NC samples due to noise in the signal introduced by the skull parts. Evaluating the influence of skull parts in the model decision was out of the scope of this work; it would take specific experiments in order to determine how influential they were in the model decision.

This paper presents how the use of Clinica and MONAI for data preprocessing and analysis facilitates reproducibility in developing a CAD tool for FTD detection. We believe that having standardized, reproducible and trustable CAD tools would ease their inclusion in patient’s clinical management practices. In fact, although thousands of AI-based tools have been developed with the potential of smoothing disease detection or outcome prediction, their real practical application is lagging [11,80,81]. The stability of Clinica methods for data preprocessing well couples with the flexibility of the procedures proposed by MONAI, setting up a highly versatile system for CAD development. We believe that such practices should be applied on data adhering to the FAIR principles when developing an AI model to study a neurodegenerative disease. Additionally, we believe that explainable AI practices such as using and interpreting attention maps should be necessary steps in building a CAD tool, in order to bring the AI realm into standard clinical practice.

## 5. Limitations

This work has some limitations. First, we used only a subset of all the images available in NIFD, in essence the T1w MPRAGE MRI at T0 (patient’s first visit). NIFD includes many more MRI modalities and other timepoints. Using all images with appropriate data management strategies could improve classification performance. Second, our CAD was built on DenseNet121, the architecture that MONAI proposes for 3D classification, without hyper-parameter tuning. It is probable that a better performance could have been yielded with different parameters and it may also be that a different architecture among those available in MONAI could have performed better on our data, yet it was outside of the scope of this work to yield max performance on the test set, as performing as good as others was enough for this application. Third, images were resized when imported in the MONAI environment; thus, possibly reducing signal quality within the MRI brain scans and finally partially hindering the model discriminative performance. Nonetheless, attention maps evaluation revealed that the model focused on gross characteristics, probably disqualifying this limitation. 

## 6. Conclusions

Although AI is now a leading technology in medical research, the real-life implementation of AI-based CAD tools in daily clinical practice is still facing obstacles. To be approved by regulators, AI-based decision support systems must be able to consistently reproduce their results on multiple sites or cohorts, while integrating with electronic health record systems. Similarly, the DL models powering the statistical engine of the CAD should be consistently updated over time to keep up with evolving clinical standards [11,80,81]. In particular, the high complexity of neurodegenerative diseases poses tough challenges for the development of reliable CAD tools, since complex diseases such as FTD, AD, and PD are characterized by a strong heterogeneity in their manifestation and underlying pathological mechanisms. Here, we presented a workflow for FTD detection based on a standardized preprocessing framework of 3D brain images, coupled with a reproducible protocol of data augmentation and Deep Learning model training and evaluation. Moreover, we used explainable AI methods to demonstrate how AI behavior can be understood by regulators and physicians. We found out that our standardized workflow for AI-based CAD tool development is comparable to other classification approaches in FTD, without compromising on reproducibility and interpretability of the DL model. Interestingly, we observed that methodological heterogeneity in FTD classification is not limited to development practices but also extends to data sources and cross-validation strategies with the latter being potentially harmful for generalizability of the results (Table 4).

Thus, we believe that health informaticians should develop AI-based CAD tools with pipeline standardization in mind, as these objectives cannot be achieved without it. In particular, we need standardized data management strategies during collection, preprocessing, and sharing, especially in case of a cross-site contribution to a database. The adherence to FAIR principles for shared data is pivotal to enhance their reusability by researchers worldwide. We believe that the widespread adoption of stronger standardization principles would foster the stability of the techniques and the reliability of findings in AI research. Within this setting, the application of explainable AI methods is required to overcome the issue posed by black box models, in particular about physicians’ trust in CAD. Increasing the understanding of AI behavior would weaken the hindering for CAD tools to be applied in real-world clinical settings, finally bringing us closer to a fruitful human–machine interaction in the biomedical field.

## Figures and Tables

**Figure 1 life-12-00947-f001:**
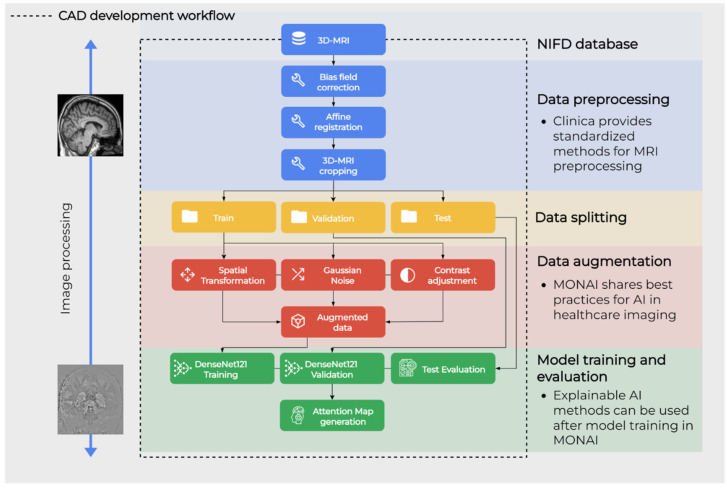
Graphical representation of the main steps of the workflow.

**Figure 2 life-12-00947-f002:**
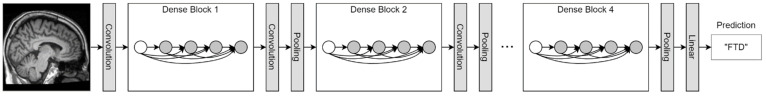
Schematic representation of DenseNet121. Its structure is similar to a classical convolutional neural network, yet DenseNet121 features dense blocks that concatenate outputs of multiple connected layers. In fact, within a dense block each layer is directly connected to every other layer in a feed-forward fashion.

**Figure 3 life-12-00947-f003:**
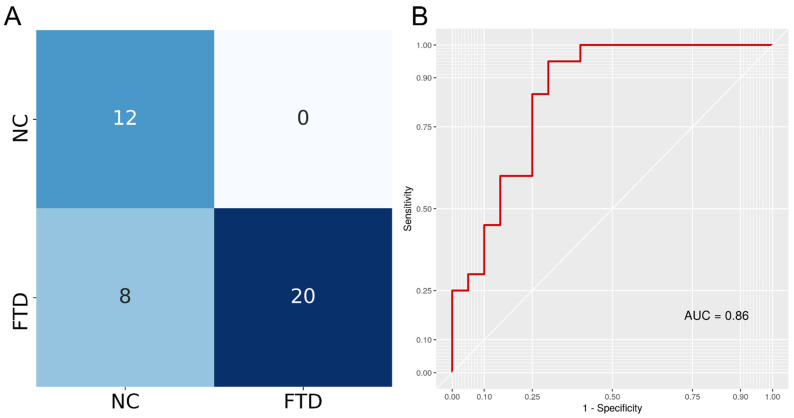
(**A**) The confusion matrix indicates classification results for both classes. Rows indicate true labels and columns indicate predicted labels. (**B**) Receiver Operating Characteristic (ROC) curve of the DenseNet121 classifier obtained when predicting disease status (FTD/NC) using 3D T1w MRI. Area Under the Curve (AUC) was calculated as the definite integral between 0 and 1 on the x-axis and provides an aggregate measure of performance.

**Figure 4 life-12-00947-f004:**
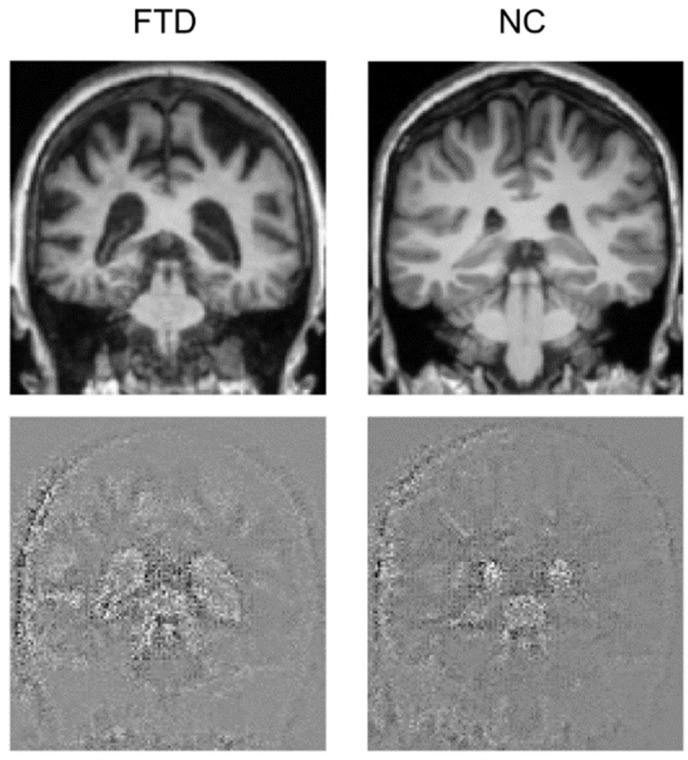
Coronal view of the brain for one FTD and one NC subject. The original brain scans used for testing are in the upper row, while the attention maps are in the lower row.

**Table 1 life-12-00947-t001:** Data splitting before augmentation.

Group	Train (n)	Validation (n)	Test (n)
FTD	143	19	20
NC	91	19	20

**Table 2 life-12-00947-t002:** Transformations applied to perform data augmentation.

Transformation	Description	Probability of Application	Specs
Translation	Translate voxels for every spatial dimension.	1	±2 voxels
Rotation	Randomly rotate the input arrays.	1	±5 degrees on the *x*-axis
Gaussian noise	Add Gaussian noise to the image.	0.5	Mean = 0; standard deviation = 2.5% of the range of activation values in the image
Contrast adjustment	Randomly updates each voxel intensity by gamma.	1	Gamma range = (0.0, 3.0)

**Table 3 life-12-00947-t003:** Train set after data augmentation.

Group	Original Images (n)	Augmented Images (n)	Total
FTD	143	257	400
NC	91	309	400
Total	234	566	800

**Table 4 life-12-00947-t004:** FTD classification results.

Citation	Comparison	Sample Size	Classification Method	Features	Metric
Proposed application	FTD vs. NC	182 FTD 130 NC	HOTS DenseNet121	3D T1 MRI scans	Acc = 0.80
Hu et al., 2020 [42]	FTD vs. NC	552 FTD 354NC	HOTS CNN	Raw 3D T1 MRI images	Acc = 0.93
Bron et al., 2017 [43]	FTD vs. NC	33 FTD 34 NC	4-fold CV SVM	Whole-brain VBM volume of GM	AUC = 0.95
Zhang et al., 2013 [44]	FTD vs. NC	25 FTD 19 NC	4-fold CV SVM	VBM GM volume on frontotemporal ROI	Acc = 0.66
Muñoz-Ruiz et al., 2012 [45]	FTD vs. NC	37 FTD 26 NC	HOTS regression	VBM GM volume	Acc = 0.85
Dukart et al., 2011 [46]	FTD vs. NC	14 FTD 13 NC	LOOCV SVM	ROIs GM	Acc = 0.85
Davatzikos et al., 2008 [36]	FTD vs. NC	12 FTD 12 NC	LOOCV SVM	PCA on RAVENS GM and WM volume	Acc = 1
Du et al., 2007 [37]	FTD vs. NC	19 FTD 23 NC	LOOCV LR	Frontal volume	Acc = 0.89
Chagué et al., 2021 [64]	FTD vs. Late Onset AD	39 FTD 34 AD	10-fold CV SVM	GM and WM volumes	Acc = 0.72
Chagué et al., 2021 [64]	FTD vs. Early Onset AD	39 FTD 34 AD	10-fold CV SVM	GM and WM volumes	Acc = 0.80
Bron et al., 2017 [43]	FTD vs. AD	33 FTD 24 AD	4-fold CV SVM	Whole-brain VBM volume of GM	AUC = 0.78
McMillan et al., 2014 [65]	FTD vs. AD	72 FTD 21 AD	HOTS linear regression	Global ventricles volume	AUC = 0.83
Dukart et al., 2011 [46]	FTD vs. AD	14 FTD 21 AD	LOOCV SVM	ROIs GM	Acc = 0.60
Lehmann et al., 2010 [66]	FTD vs. AD	23 FTD 17 AD	CV SVM	Whole brain cortical thickness	Acc = 0.79
Davatzikos et al., 2008 [36]	FTD vs. AD	12 FTD 37 AD	LOOCV SVM	PCA on RAVENS GM and WM volume	Acc = 0.84
Klöppel et al., 2008 [67]	FTD vs. AD	19 FTD 18 AD	LOOCV SVM	GM volume	Acc = 0.89

Acc: Accuracy; AUC: Area Under the Curve; CV: Cross-Validation; GM: Grey Matter; HOTS: held-out test set; LOOCV: Leave-One-Out Cross-Validation; LR: Logistic Regression; SVM: Support Vector Machine; VBM: Voxel-Based Morphometry; WM: White Matter. In the case that a paper presents multiple classification results based on different feature sets (e.g., comparing classification performance on hippocampus volume vs. VBM-GM volume), only the best result was reported in this table. Moreover, only results obtained with brain morphometry data were considered. Accuracy was reported where available, otherwise AUC was reported. The table is arranged by Comparison (descending) and year of publication (descending).

## Data Availability

This study was performed using the NIFD. See the acknowledgements.

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
