# Peer review of "A Reproducible Deep-Learning-Based Computer-Aided Diagnosis Tool for Frontotemporal Dementia Using MONAI and Clinica Frameworks"

_life, 2022, doi:10.3390/life12070947_

Round 1

Reviewer 1 Report

This study used explainable AI methods to classify FTD from controls using a reproducible approach and publicly available data. They found a comparable performance to previous studies, and showed that the classifier relied on the ventricles and skull. Several previous studies have attempted to classify FTD from controls using neuroimaging data before. Nonetheless, the methods are sound and described in detail. The article has minor grammar mistakes and unconventional English use throughout.

The authors found that the skull played a large role in the classifier’s functioning. They later say in the discussion that this may have been why the NC cases were misclassified. Can the authors explain why they think the skull was included, and perhaps give any clinical reasoning or evidence as to why this might happen?

From an image analysis perspective, it is easy to separate skull from brain tissue, and so I have to ask why the authors did not first use brain-extracted images as their primary input. Was there a prior reason to suspect that the non-brain tissues could aid in the classification?

Reference 31 is not appropriate. It is a review. Please instead cite the original article showing perfect classification.

Please check the references for the formatting of names of initiatives e.g. ADNI in reference 32 appears as “Initiative, A.D.N.”, instead of the full name.

The authors include detailed descriptions for various tools and methods they used in the methods section. These include Clinica, MONAI, NIFT, FTLDNI, attention maps and DL. These descriptions should instead be placed in the introduction. They are also too long, containing a lot of extraneous information.

The footnote to Table 1 (line 203) is repetitious with the text and should be removed.

Reviewer 2 Report

·       The abstract should be revised to show clearly the problem statements with brief review of the current work. Throughout this review, highlight why the reproducible is needed based on your findings.

·       In addition, you should discuss the methodology, and then the results. Besides, the following sentence should be rewritten to enhance the clarity.  sentence “The DL model achieved a performance comparable to 22 other FTD classification approaches, yielding .80 accuracy (95% confidence intervals: .64, .91), 1 sensitivity, .6 specificity, .83 F1-score and .86 AUC.”

·       The introduction is informative but needs to be restructured to show the significance of the work. For instance, the first part, which talks about CAD, can be minimized. Discuss deeply why DL is advantageous. Discuss the states of art of this field. Then highlight your problem statement, research objectives, and contributions.

·       The paper lacks the critical review of related works and state-of-the-art of this filed, which affect the research soundness.

·       Figure one discusses the used methodology, however it is shallow. I can’t see in the figure  the role of DL, NOMAI, and Clinica; and also the different stages of the model: preprocessing, training, testing, etc.   The figure should be specific in showing all the methods used in sequence. Another point is the figure is misplaced. It should be discussed and cited in the main text.

·       In the section of dataset description: “2.1. Neuroimaging in Frontotemporal Dementia database”. The dataset should be specified in terms of name and source, what are the main features of the dataset, do you use all the features. Did you analyze the features correlations, etc. Please discuss deeply. Did you apply any filtration techniques on the dataset? Please explain.  Another point is the section 2.4 and 2.5 should come after 2.1.

·       Please check the section numbering, as well.

·       In line 197, justify your selection of data segmentation 70/30.

·       In the attention map, how did you calculate the attention rate, and how is it used to find the relevant areas?

·        Section 3.2 and section 2.7 have the same name, which is misleading. Please check the naming or combine both sections in one.

·       Improve the figure 2 and discuss the confusion matrix in details with relevance to your work.

·       Deep learning model should be elaborated to show its structure, and parameters tuning.

·       The results should be improved to show different performance metrics, as well as, show the comparison of your work with similar works based on the literature review.

·        The conclusion should include a critical review of the results and findings of the author's research. This part should be revised.

·        

Round 2

Reviewer 1 Report

Thanks, the authors have addressed my questions sufficiently.

Reviewer 2 Report

I have no further concerns on this paper. the authors did well to address all my previous concerns. Thanks for them.